# Experimental and Numerical Investigations of Cement Bonding Properties

**DOI:** 10.3390/ma14237235

**Published:** 2021-11-26

**Authors:** Ionut Lambrescu, Catalin Teodoriu, Mahmood Amani

**Affiliations:** 1Department ITIMF, Petroleum-Gas University Ploiesti, 100680 Ploiesti, Romania; ilambrescu@upg-ploiesti.ro; 2Mewbourne School of Petroleum and Geological Engineering, The University of Oklahoma, Norman, OK 73019, USA; 3Petroleum Engineering Program, Texas A&M University at Qatar, Doha P.O. Box 23874, Qatar; mahmood.amani@qatar.tamu.edu

**Keywords:** well construction, well integrity, well cementing, interfacial bonding strength of cement

## Abstract

Well integrity is of high importance during the entire well life span especially when renewable energy resources such as geothermal are designed to cover the increasing world energy demand. Many studies have documented the importance of the casing–cement interfacial bonding to ensure critical well integrity achievements; however, laboratory experiments and field data are not always aligned. Furthermore, Finite Element Analysis shows relatively high discrepancies compared with the results of various scholarly published works. The limitations in the FEA are most probably generated by the casing–cement interaction modeling parameters. Typically, the contact between casing and cement is modeled using the so-called CZM method, which includes the shear debonding process into the FEA. Several setups have been used in the past to determine the interfacial casing–cement bonding shear strength. Some of these setups are briefly summarized herein. The novelty of this paper consists in the combination of a relatively simple experimental setup with the finite element modeling of the experiment itself to demonstrate that it is important to acquire accurate laboratory data for debonding simulations and, thus, to improve the well integrity prediction. The aim of this paper is to better understand the limitations of the finite element method when modeling shear bonding of the cement and, in the same, to verify that the proposed experimental setup can be modelled using numerical approaches. The successful numerical simulation can later be used for upscaled models. The results confirm the experimental push down setup and aid engineers to further understand and validate CZM models and optimize the well design to achieve maximum well integrity potential. Our results are within 1% error from the average field data.

## 1. Introduction

Wellbore integrity has many facets, but the main element of failure remains the system casing–cement. Wellbore cement is commonly used to seal the annular space between casing and formation through mechanical and hydraulic seals [1]. The ability of the cement to maintain good properties during the life of the well is very important and has been strongly documented by various authors [1,2,3,4].

The very first documentation of cement bonding shear strength was provided by Evans [5], who reported experimental results of so-called bonding strength and hydraulic bonding strength. While the hydraulic bonding strength is a much more complex property of the cement that shows the casing–cement system’s ability to seal, the mechanical bonding strength or the Interfacial Bonding Shear Strength (IBSS, as it will be referred to in this paper, is a measure of how strong the casing and the cement are stuck together (bonded)). This bond can fail under two specific load types: perpendicular to the bonded surface (tensile bonding) and along the bonded surface (shear bonding). The recent literature overview shows an intensive focus on the casing–cement–formation interaction. Geothermal wells, for example, are critically impacted by the relative casing–cement motion that requires a detailed understanding of the cement IBSS [6,7]. Long-term cement properties also play an important role since these properties suffer continuous changes. The lack of accurate information about long-term cement properties makes the finite element study of a well situation very difficult [8]. The IBSS is also affected by the cement ageing [9].

Typical IBSS testing consists of two methods: push-out and rotational disk. The push-out is the most commonly used version, and it is a development based on Evan’s [5] initial experimental work. [9,10,11] have also shown the importance of proper testing methods and data evaluation in order to perform finite element studies. Several authors have published results obtained using push out setup or variations of it [9,12,13,14,15,16,17,18]. A different approach was taken by [10], which uses a modified Brazilian test to measure the mixed-mode interfacial strength.

Previously published data on the casing–cement interfacial bonding shear strength vary from author to author [12,13,14,15,16,17,18]. This variation is mainly due to the testing method used to record the bonding shear strength but also to the sample curing time and cement slurry composition. However, the published data is limited and allows only a qualitative comparison. For example, it seems to be very common to measure the interfacial bonding shear strength after one day (24 h) of curing. Table 1 shows some of the reported 1-day bonding shear strength at room temperature.

Given the scarcity of finite element studies that involved real experimental data for IBSS, this paper proposes investigating how far and accurately the CZM method can model experimental tests for cement IBSS. Our main hypothesis is that by using the measured experimental shear strength, the numerical solution should lead to the same push-out force as for the experimental results. Furthermore, our aim is to investigate CZM parameters’ sensitivity to the proposed simulation.

## 2. Experimental Approach

This paper shows the numerical and experimental results performed in order to better understand the casing–cement bonding behavior and especially to what extent modern numerical simulators allow the mimicking of experimental work. Firstly, we describe the experimental methodology used in this work, starting with sample preparation and a full description of the sample geometry. Secondly, we present the numerical approach used to model the experiment. The specimen size and type used for this work are similar to those proposed by [9].

### 2.1. Sample Preparation

For this work, we used Class H neat cement with micro silica as an additive. The cement mixing was performed according to [19,20]. The entire equipment used for mixing is API certified and was explained in detail by Teodoriu et al. [9]. Water and cement with a specific ratio of 0.38 have been mixed using a certified API mixer for 50 s. The resulted slurry was poured into cube molds (for UCS measurements) and the cylindrical cement shear bonding specimens. All shear bonding specimens are manufactured from 304 Stainless Steel, and the inner surface was fully polished with 400-grid sandpaper. The pouring took about one minute, after which all samples were gently immersed in the water bath. All tests presented in this paper were performed at room temperature. Therefore, the water bath was not started. Distilled water was used for mixing and curing the samples. No pressure was applied on the samples; hence, all tests were performed at atmospheric pressure. The samples were cured for 1, 3 and 7 days. The samples were tested immediately after being extracted from the water bath.

### 2.2. Experimental Setup

A detailed description of the experimental setup used for this work can be found in [9]. The cells (molds) utilized for the bonding experiments presented in this paper are shown in Figure 1.

Figure 2 shows a cross-sectional view of the interfacial bonding shear strength cell. The IBSS cell utilized for these experiments uses the cement inside of a steel cylinder, thus providing a very precise and easy-to-control contact area. The inside of the pipe is finely polished using 400-grid sandpaper. The push-out rod used for the experiments has a slightly smaller diameter than the steel cylinder ID. It is designed to fully cover the cement surface and to allow a uniform push-out force on the cement. The load and displacement change during the experiment was measured and recorded with a sampling rate of 10 Hz and was used for numerical analysis as well.

Table 2 shows the detailed geometry of the shear bonding cell, whereas the length, outer diameter and inner diameter (ID) are average values based on 4 measurements. The values presented in Table 2 are used for the numerical study, as described in the next section, and to calculate the interfacial bonding shear strength using Equation (1).

The interfacial bonding shear strength (measured in MPa) is calculated as:(1)$$\sigma =\frac{F_{max}}{2\pi *ID_{A}*CL}$$
where *F_max_* is the maximum recorded force, N;*ID_A_* is the inner diameter of the cell, m;*CL* is the interfacial bonding shear strength cell length, m.

An example of the piston force versus piston displacement is shown in Figure 3. The maximum force and its corresponding relative displacement is extracted from this graph.

Table 3 shows the calculated interfacial bonding shear strength for 1, 3 and 7 days, based on average maximum recorded force values of 4 samples. The common trend of the IBSS increasing with curing time is obvious.

## 3. Numerical Approach

The numerical study was carried out using the 2021 R2 ANSYS version. In order to conduct this study, we created an axis symmetric model of the interfacial shear bonding cell using the average geometry data shown in Table 2. Figure 4 shows the model used for this study.

### 3.1. Element Selection and Definition of Contact

Nonlinear mechanical, quadratic axisymmetric elements with three or four edges were used for the whole cell in order to obtain better results. Figure 5 presents the meshed assembly. As it can be seen, the mesh is strongly refined in the contact areas using edge sizing conditions. The average element size is 2 mm (imposed by a face sizing condition). In the cement–pipe contact area, the average element size is 0.05 mm, while in the piston–cement contact area, the average element size is 0.5 mm. This refinement allows an accurate contact force and contact process simulation.

The model exhibits two contact zones: one between the piston and the cement and another one between the cement and the pipe interior wall. Both were modeled as bonded, but for different reasons.

The contact between piston and cement is not rigorously bonded since this area is not very important for our study, and for simplifying the model, the authors reckoned that a bonded contact will be enough.

The cement-pipe internal wall contact had to be modeled as bonded, since a CZM (Cohesive Zone Material) will be inserted in this area. In this case, Ansys imposes that the supporting contact be bonded. For this contact, the Pure Penalty formulation was used. From the available CZM materials, the authors chose the bilinear one and the pure Mode II debonding model, as presented in Figure 6.

The mathematics [21] behind the considered model is described in the equation below:(2)$$T_{t}=K_{t}\delta_{t}\left(1-D_{t}\right)$$
where:

$K_{t}=\frac{T_{t}^{max}}{\delta_{t}^{*}}$ is the tangential cohesive stiffness;$T_{t}^{max}$ is the maximum tangential cohesive traction;$\delta_{t}^{*}$ is the tangential displacement jump at maximum tangential cohesive traction;$\delta_{t}^{c}$ is the tangential displacement jump at the completion of debonding;$\delta_{t}^{max}$ is the maximum tangential displacement jump attained in deformation history;$D_{t}$ is the damage parameter associated with Mode II/III dominated bilinear cohesive law:(3)$$D_{t}=\left\{\begin{array}{c} 0,\;\;\;\delta_{t}^{max}\leq \delta_{t}^{*}\\ \left(\frac{\delta_{t}^{max}-\delta_{t}^{*}}{\delta_{t}^{max}}\right)\left(\frac{\delta_{t}^{c}}{\delta_{t}^{*}-\delta_{t}^{*}}\right),\;\;\delta_{t}^{*}<\delta_{t}^{max}\leq \;\delta_{t}^{c}\\ 1,\;\;\;\delta_{t}^{max}>\delta_{t}^{c} \end{array}\right.$$

Only three parameters are set in the Engineering Data section in Ansys for the considered case:T_t_^max^—Maximum Equivalent Tangential Contact Debonding (calculated from the experiments for each case);δ_t_*—Tangential Slip at the Completion of Debonding, also calculated from the experiments for each case;Artificial Damping Coefficient. This parameter is intended to assure the convergence of the analysis and, as a rule of thumb, should be smaller than the smallest increment time step (as an analysis parameter).

### 3.2. Load and Boundary Description

The outer steel cylinder, as can be seen in Figure 2, is simply supported on the inferior supporting face. To simplify the model, the supporting pipe element was eliminated (so there is no need for another contact region), and the inferior horizontal face of the pipe was blocked (fixed). As already mentioned, the contact between the push-out rod and cement was selected as fully bonded, but this option does not have any effect on the results. The load is ramped applied in a manner that mimics the way the force varied during the experiment up to values large enough to induce instability in the model (that becomes unstable) and thus allows detecting when the debonding process initiates. This will be described in detail in the next chapter.

The cement and steel relevant properties used for this numerical analysis are shown in Table 4. Analyses for different values for the Young’s modulus, respectively, and Poisson ratio of the cement were performed, and the results will be presented later in this paper.

Figure 7 shows the full model with load and constrains applied. The force is applied on the upper piston, which is modeled out of steel in order to create a uniform push-out pressure on the cement top. The contact element between the piston and the cement is modeled as compression only. The piston is constrained to move in a vertical direction only. The outer cylinder made out of steel is modeled as being fully supported at the bottom (fixed support). Once debonding force is achieved, the cement is free to move downwards. The cement is free to move in any direction; however, it is initially “kept in place” by the bonding elements between cement and casing.

### 3.3. Failure Mode Definition

We used the following procedure to identify the debonding starting point. The force is applied in small steps until the maximum applied value is reached. The maximum applied force used for the simulation is actually any force that will exceed the experimentally measured one, typically about 125% of the measured forced. The simulation log file is revisited, and the graph force versus displacement is plotted. The point at which the cement debonding is considered to initiate is found observing discontinuities in the plots of various results. Figure 8, Figure 9, Figure 10 and Figure 11 illustrate the case of 7-day cement.

Figure 8, Figure 9, Figure 10 and Figure 11 show that all analyzed results lead to the same value for the starting of the debonding process. The four different ways to identify the moment in time when the contact elements are losing the contact indicate the debonding process start: 299.11, 299.1, 299.04 and 299.1 s, respectively. Since the force is applied with a linear variation of 324 s and a maximum value of 31,000 N (for the 7 days case), the forces for starting debonding are: 2,861.655, 286,176, 2,861.185 and 286,176 N, respectively. The four values are distributed on a 4.7 N interval that is 0.016% of the maximal value.

## 4. Results and Discussions

Figure 12 and Figure 13 show the evolution of Force versus Displacement for the samples after 1 and 7 days. The curves were used to extract the maximum recorded force and the corresponding displacement. The values were utilized in the FEM analysis as follows: the displacement was used as an input variable to fully define the CZM model. Two methods were applied to compute the CZM parameter called tangential slip. The first method uses the total recorded displacement of the cement during the experiment. Since the cement is pushed out continuously, also after the debonding took place, this method might lead to errors, and thus, a second method was proposed. The second method is computing the tangential slip as being six times the measured displacement occurred at maximum recorded force to push out the cement. As shown in Table 5, there is little difference between the two methods, which indicates the Maximum Equivalent Tangential Contact Debonding as the main parameter for the CZM model.

The maximum measured force was compared with the maximum force obtained from the FEM. As presented above, the failure mode was defined as the point when different output results (axial displacement, contact status, reaction force in the contact or contact sliding) exhibit an abrupt change in their values. Table 5 shows the measured and calculated force to debond the cement for 1 day, 3 days and 7 days cement.

The general debonding behavior can be observed by comparing the force versus displacement charts shown in Figure 12 and Figure 13. The one-day curing samples have shown a relatively smooth behavior with a low maximum measured push-out force. On the other hand, the 7-day cured samples have shown a strong debonding process with jumps and an overall erratic push-out motion. However, the overall maximum recorded forces for all six samples have shown a good consistency despite all the erratic motion. This means that only the post debonding process has erratic behavior, but the measured maximum bonding force is accurate. Further investigations are ongoing currently to understand this phenomenon.

A series of numerical calculations was generated to determine the sensitivity of the FEM analysis to the input variables, especially the cement’s Young’s Modulus and Poisson Ratio. The Young’s Modulus was varied from 4137 MPa to 20,685 MPa, as found in the literature [9,10,11,12,22]. Additionally, a very low Young’s Modulus (500 MPa) was used in order to verify the stability of the numerical method. A very low Young’s Modulus will be the equivalent of the cement not being fully set. Figure 14 shows the sensitivity of the calculated debonding force with change in Young’s Modulus for 1 Day curing for three Poison ratio values: 0.2, 0.3, respectively 0.4.

It can be seen that the numerical calculation response shows almost no reaction to the change of Young’s Modulus. A rescaled view of the same graph as shown in Figure 15 shows that the only minor change is visible at Young’s Modulus value of 500 MPa from 8793 to 8700 N.

Additionally, we noticed that the Poisson Ratio does not influence the bonding calculations at all. However, it must be noticed that the results seem to overlap with the change in Poisson Ratio, see Figure 16.

This effect was not noticed for the numerical simulations for the 7 days curing time—see Figure 17 and Figure 18 for Young’s Modulus Sensitivity and Figure 19 for Poisson Ratio sensitivity. Figure 17 shows the sensitivity of the calculated debonding force with change in Young’s Modulus for 7 Day curing for three Poison ratio values: 0.2, 0.3, respectively 0.4. A zoom of the data shown in Figure 17 is presented in Figure 18, revealing just little differentiation between the three cases. On the other hand the Figure 19 shows that Young modulus shows some clear effect on the results.

A total number of 64 simulations were performed in order to obtain the sensitivity data shown in this paper.

Basically, our simulations show that the bonding properties do not vary much as a function of Young’s Modulus and Poisson Ratio. Nevertheless, we noticed much more organized results for the 7-day curing simulations, which leads to the idea that the most important factors affecting the results are the CZM parameters, especially the Maximum Equivalent Tangential Contact Debonding Parameter, which was calculated from the experimental data.

## 5. Conclusions

The paper demonstrates that accurate experimental design and finite element analysis can simulate casing–cement debonding. The sensitivity study has shown a low effect of the Poisson Ratio and Young’s Modulus on the overall debonding force.

The main parameters that affect the debonding process are the IBSS and the use of the CZM modeling for the bonding elements. However, the Maximum Equivalent Tangential Contact Debonding plays the most important role in the results of FEA, and thus, we have shown how to obtain this value experimentally.

The experimental results were well modeled by the FEA with acceptable errors under 1%.

The main limitation of the presented study lies in the fact that the numerical simulator cannot yet model the post debonding behavior, which we believe is essential for real situations.

## Figures and Tables

**Figure 1 materials-14-07235-f001:**
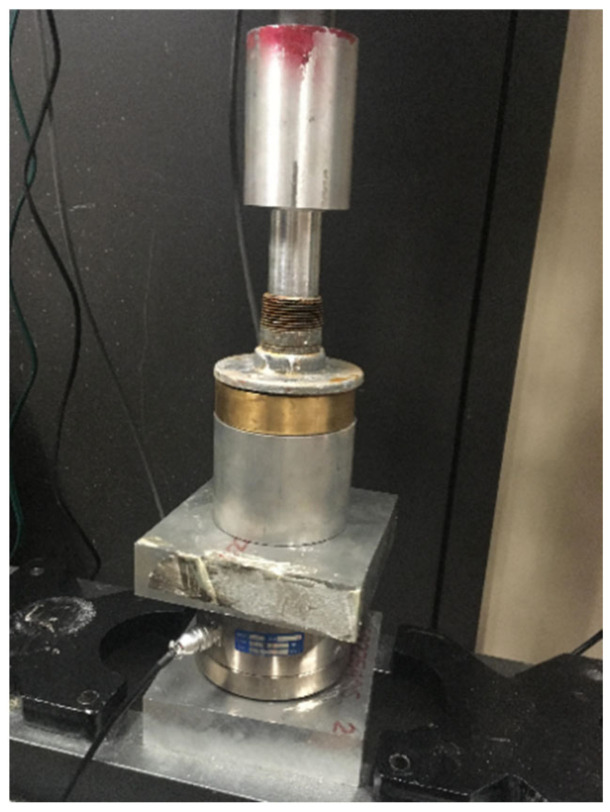
Cement shear bonding cell used for this work.

**Figure 2 materials-14-07235-f002:**
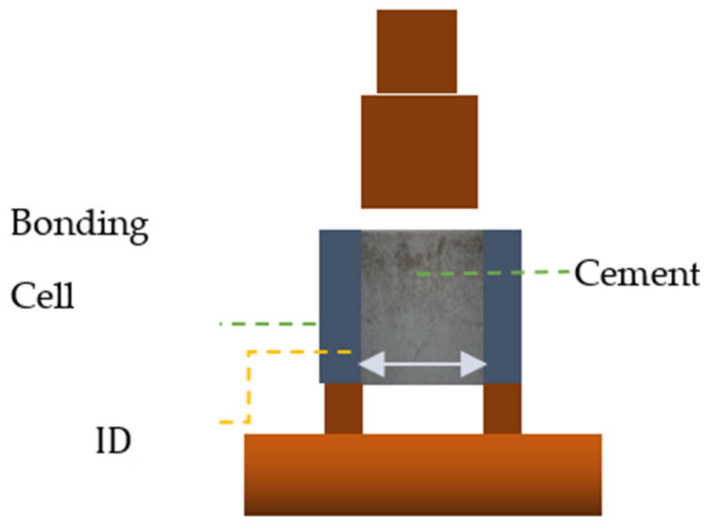
Interfacial Shear Bonding Strength test cells used in the current study.

**Figure 3 materials-14-07235-f003:**
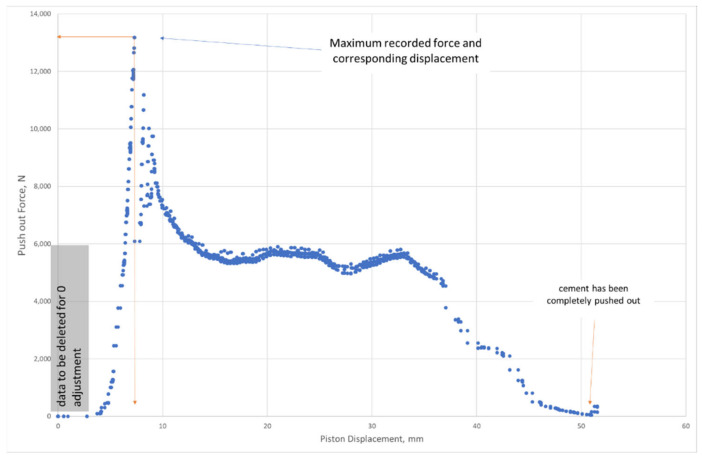
Recorded push-out force versus piston displacement used to extract and calculate the interfacial bonding shear strength.

**Figure 4 materials-14-07235-f004:**
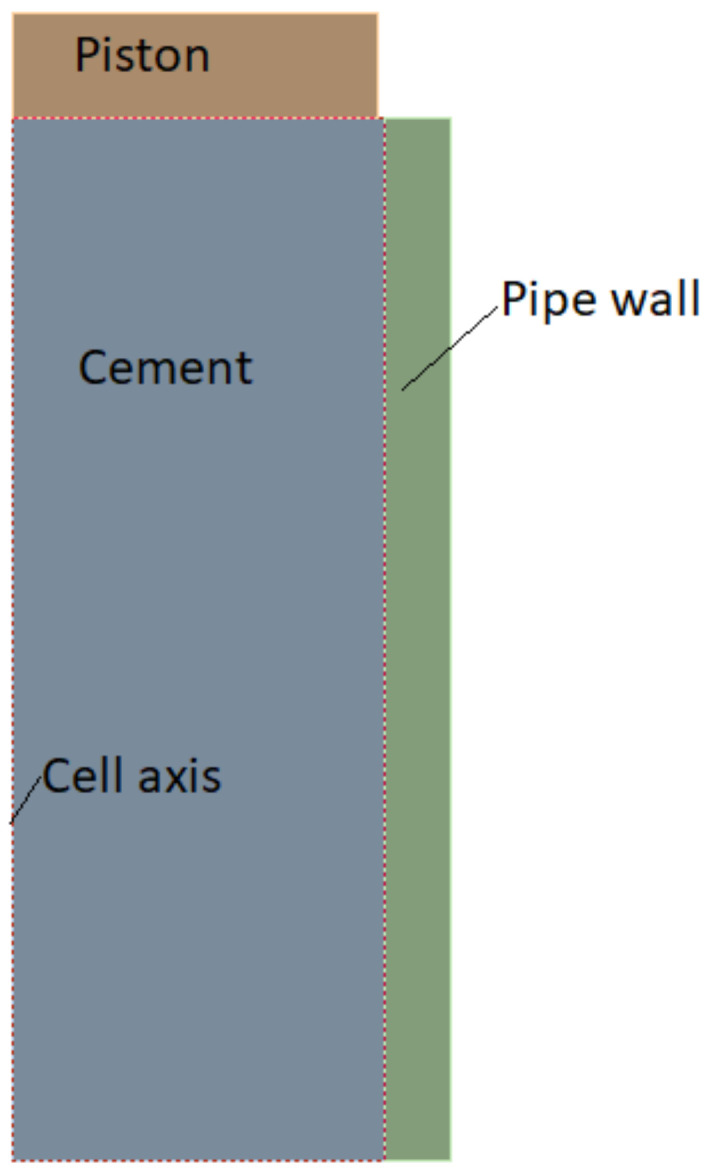
Model used for the study.

**Figure 5 materials-14-07235-f005:**
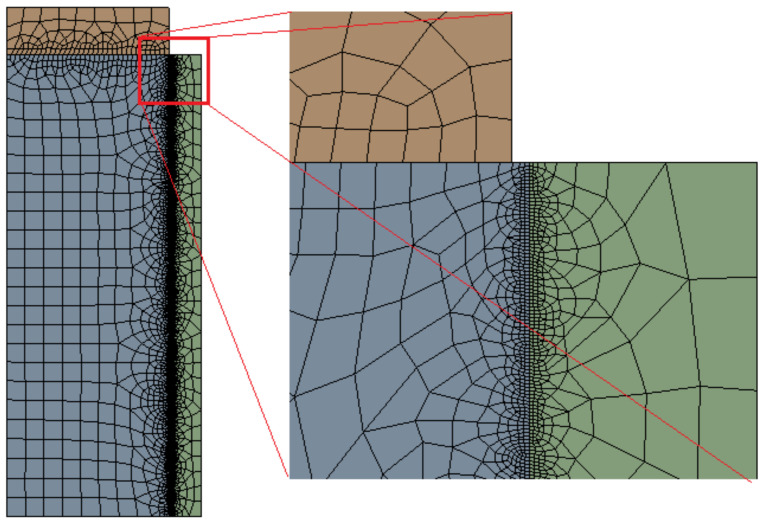
Meshed assembly of the experimental cell, showing the element density near to bonded surface.

**Figure 6 materials-14-07235-f006:**
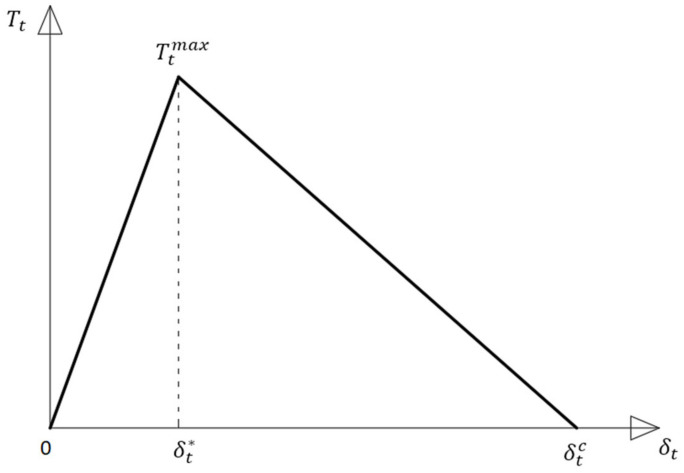
The bilinear, Mode II model.

**Figure 7 materials-14-07235-f007:**
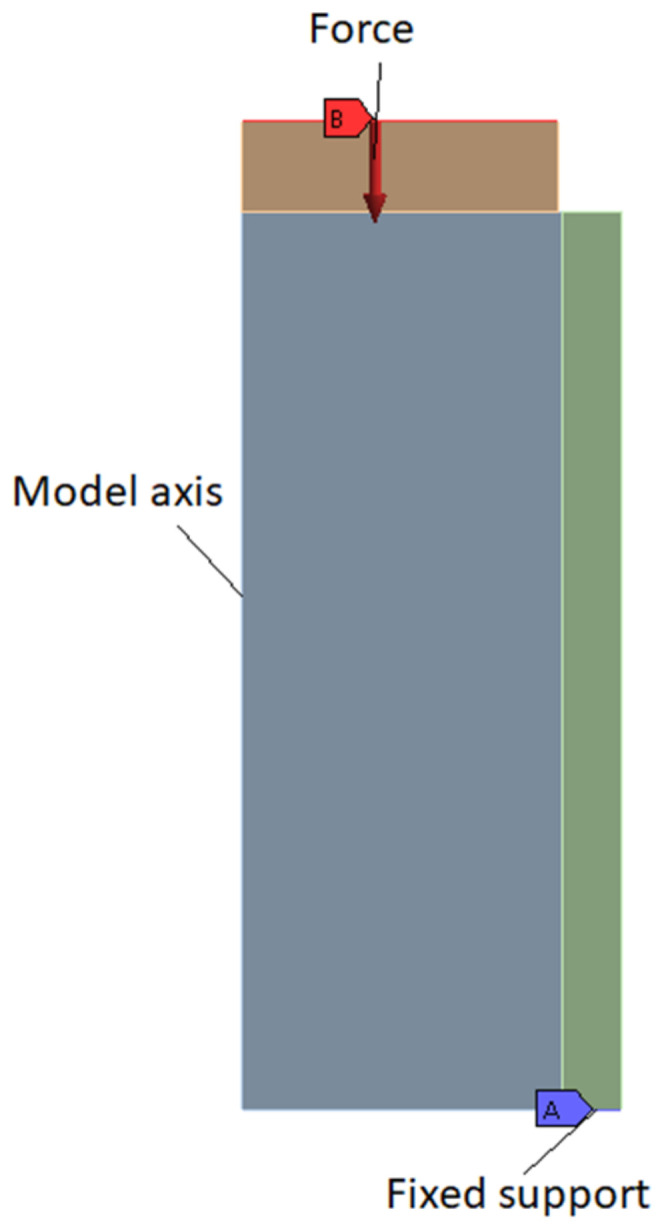
Model with boundary conditions and loads.

**Figure 8 materials-14-07235-f008:**
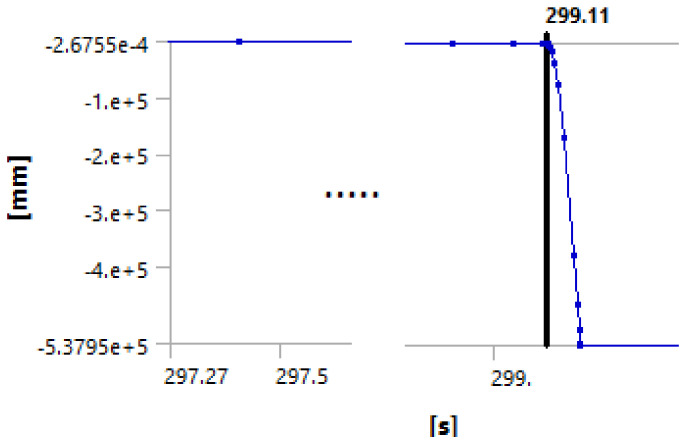
Debonding moment observed on the axial displacement plot showing abrupt change of displacement by the blue line at time stamp 299.11 s.

**Figure 9 materials-14-07235-f009:**
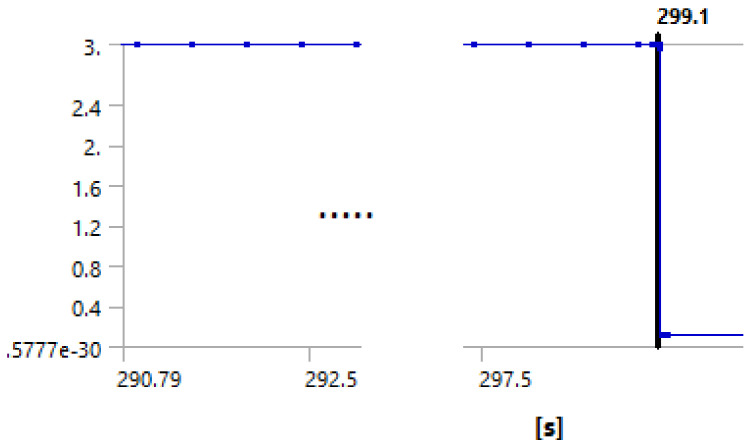
Debonding moment observed on the contact status showing abrupt change of contact status by the blue line at time stamp 299.1 s.

**Figure 10 materials-14-07235-f010:**
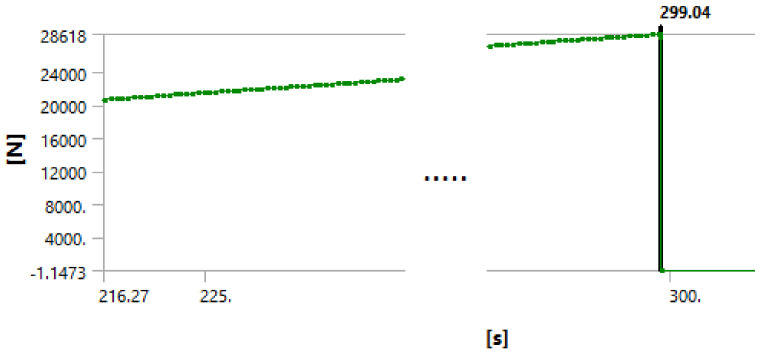
Debonding moment observed on the reaction force in the piston showing abrupt change of piston force by the green line at time stamp 299.04 s.

**Figure 11 materials-14-07235-f011:**
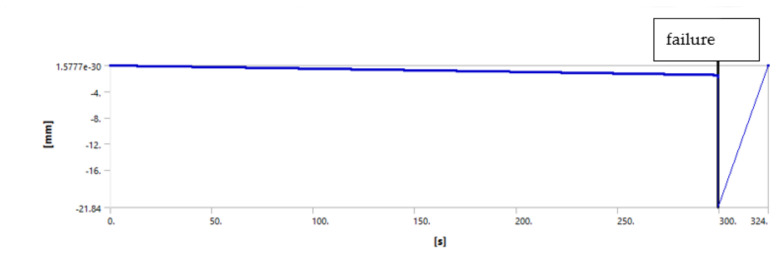
Debonding moment observed on the contact sliding distance showing abrupt change of sliding distance by the blue line at time stamp 299.1 s.

**Figure 12 materials-14-07235-f012:**
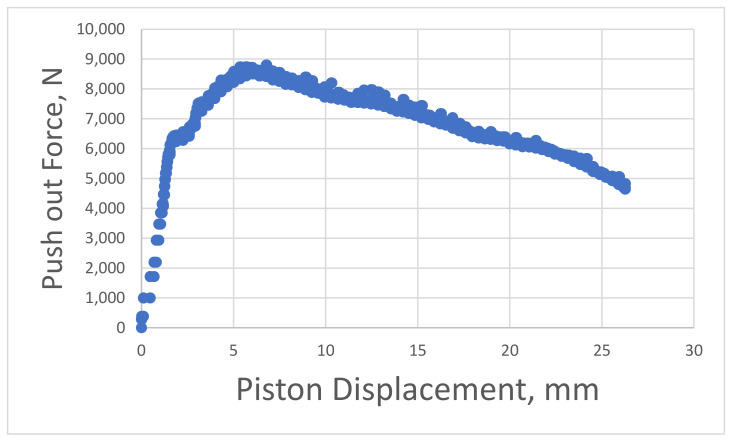
Push-out force versus piston displacement for sample cured for 1 day.

**Figure 13 materials-14-07235-f013:**
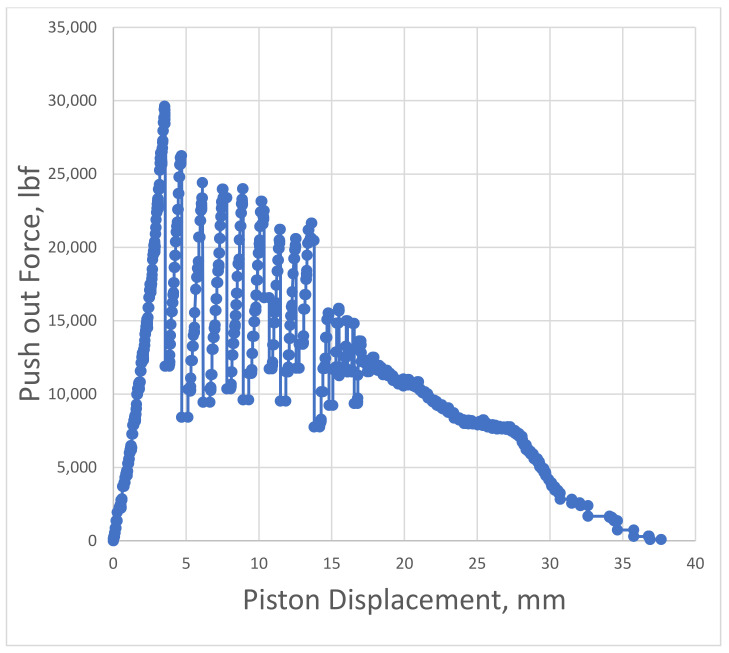
Push-out force versus piston displacement for sample cured for 7 days.

**Figure 14 materials-14-07235-f014:**
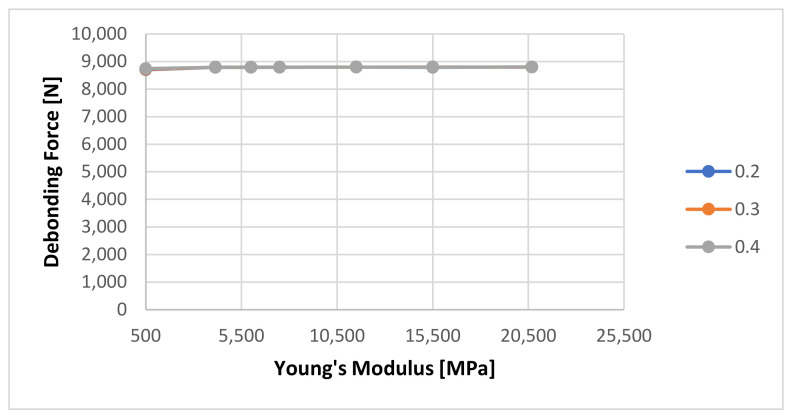
Calculated debonding force as a function of Young’s Modulus of the cement with a 1-day curing time, please note that all three lines are overlapping showing just one gray line.

**Figure 15 materials-14-07235-f015:**
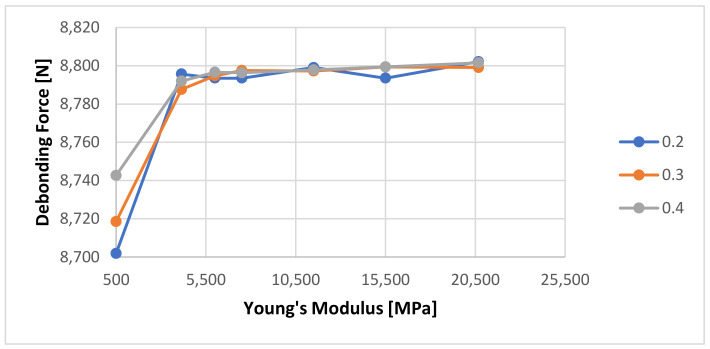
Detailed view of the calculated debonding force as a function of Young’s Modulus of the cement with a 1-day curing time.

**Figure 16 materials-14-07235-f016:**
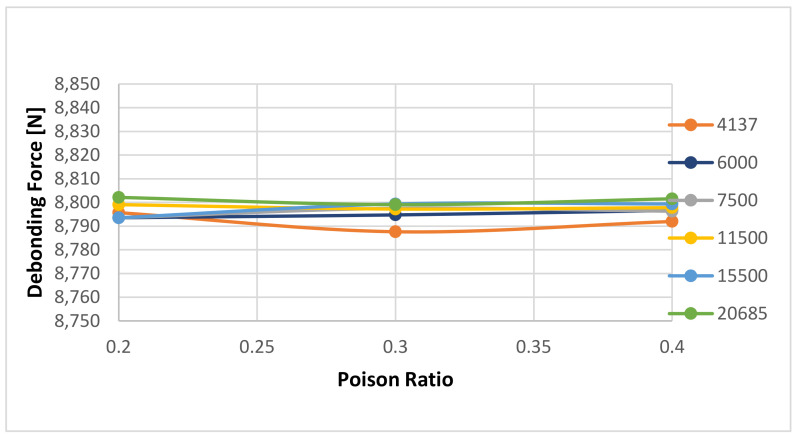
Detailed view of the calculated debonding force as a function of Poisson Ratio and selected Young’s Modulus of the cement with a 1-day curing time.

**Figure 17 materials-14-07235-f017:**
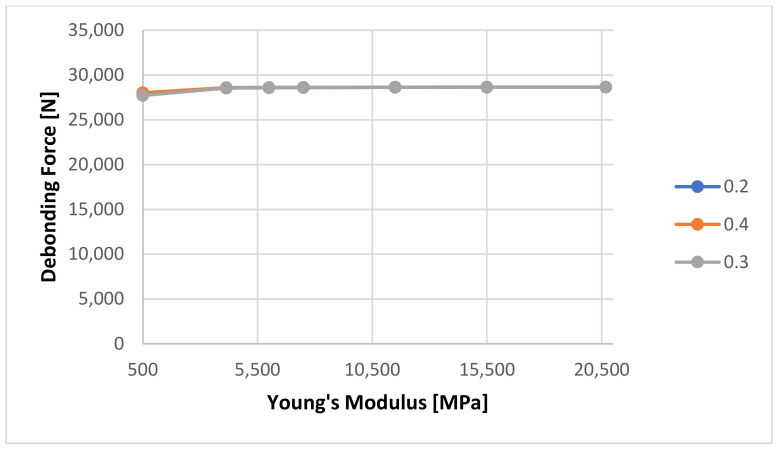
Calculated debonding force as a function of Young’s Modulus of the cement for 7 days curing time, please note that all three lines are overlapping showing just one gray line.

**Figure 18 materials-14-07235-f018:**
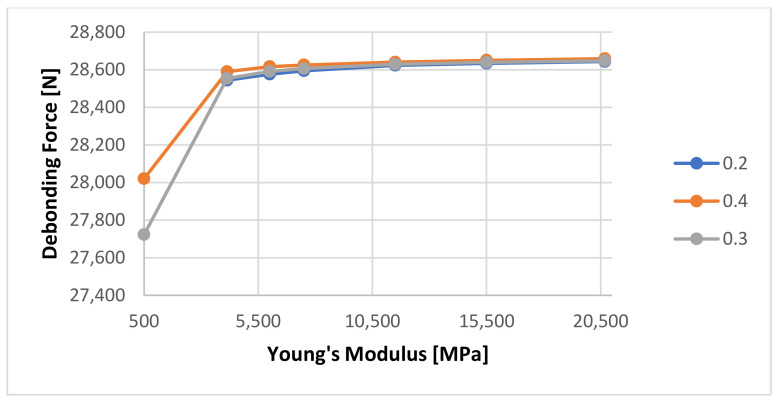
Detailed view of the calculated debonding force as a function of Young’s Modulus of the cement with a 7-day curing time.

**Figure 19 materials-14-07235-f019:**
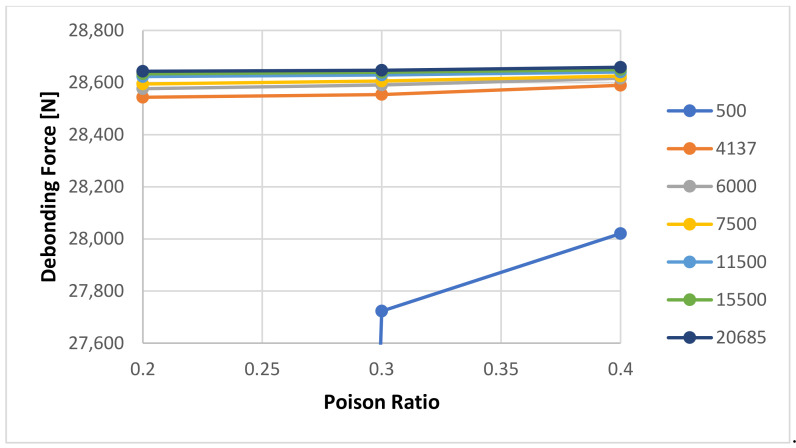
Detailed view of the calculated debonding force as a function of Poisson Ratio and selected Young’s Modulus of the cement with a 7-day curing time.

**Table 1 materials-14-07235-t001:** Comparison and validation of the preliminary results of this work with values in published literature.

Author	Salehi et al. 2016 [13]	Lavrov and Torsaer 2016 [14]	Zhao et al. 2015 [12]	Zhao et al. 2015 [12]	Teodoriu et al. 2018 [9]	Tabatabei et al. 2020 [10]
Comment	After 24 h	-	After 5 days different temp.	Added sand to casing	After 24 h	After 24 h
Interfacial Bonding Shear Strength (PSI)	81	14.5 to 145	14.5 to 362	362 to 1090	68–300	40
Interfacial Bonding Shear Strength (MPa)	0.56	0.1 to 1.0	1.0 to 2.5	2.5 to 7.5	0.47–1.94	0.271

**Table 2 materials-14-07235-t002:** Geometries of the shear and bonding cells.

Item	Cell Length (mm)	Outer Diameter (mm)	Inner Diameter (ID_A_) (mm)
Bonding Cell	50	40	35.1

**Table 3 materials-14-07235-t003:** IBSS for the cement samples cured at 1 day, 3 days and 7 days.

Curing Time [Days]	Measured Force [N]	Calculated IBSS [MPa]
1	8795	1.60
3	13,180	2.40
7	29,616	5.10

**Table 4 materials-14-07235-t004:** Cement and steel properties.

Material	Young’s Modulus (MPa)	Poisson Ratio
Cement	4137–21,000	0.2–0.4
Steel	210,000	0.3

**Table 5 materials-14-07235-t005:** Measured and calculated forces for the three simulated cases 1 day, 3 days and 7 days.

Curing Days	Measured Force [N]	Calculated Force Using 2 Methods [N]	Absolute and Relative Error for the 2 Proposed Methods
1	8794	8792/8793	−1.1(−0.012%)/−0.17(−0.001%)
3	13,180	13,179/13,170	−0.98(−0.007%)/−9.54(−0.07%)
7	29,590	28,618/28,606	−972.13(−3.28%)/−983.88(−3.32%)

## Data Availability

Not applicable.

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
