# Peer review of "Experimental and Numerical Investigations of Cement Bonding Properties"

_materials, 2021, doi:10.3390/ma14237235_

Round 1
Reviewer 1 Report
Manuscript #1439244 reports an exciting approach to the casing-cement bonding interface commonly used in other fields and smaller dimensions. Some minor corrections could be performed to improve the manuscript.
- Perhaps, the term “numerical” could be replaced with “quantitative”.
- What is the cement´s total setting time? What is the cement´s initial setting time?
- What is the cohesive strength of the cement? Was it considered to calculate the shear strength?
- How was the sample size (n = 4) calculated (line 115)?
- What was the load cell of the testing machine?
- During the 3 and 7 days of testing, what were the samples´ storage conditions?
- What type of bond is formed between the cement and the well?
- The authors mentioned that “The force is applied in small steps until the maximum applied value is reached” (lines 199-200) – what is the crosshead speed?
- What was the casing thickness? Do the authors think that casing thickness could affect the results?
- How rough was the internal casing surface? Could the internal roughness affect the interface bond shear strength?
Author Response
Reviewer 1:
Manuscript #1439244 reports an exciting approach to the casing-cement bonding interface commonly used in other fields and smaller dimensions. Some minor corrections could be performed to improve the manuscript.
- Perhaps, the term “numerical” could be replaced with “quantitative”.
Answer: Thank you for your suggestion. However, the method used in this paper is based on finite element method which considered a numerical approach. We have however explained this in much detail in the introduction.
- What is the cement´s total setting time? What is the cement´s initial setting time?
- Answer: the cement setting time was 1, 3 or 7 days as explained in the methods sections. We have added one more sentence to make sure this is well explained. The initial and total setting time are the same, hence 1, 3 or 7 days.
- What is the cohesive strength of the cement? Was it considered to calculate the shear strength?
Answer: we din not measured the cohesive strength. Our experimental measurements allow a direct calculation of the shear strength.
- How was the sample size (n = 4) calculated (line 115)?
Answer: According to API 3 to 6 samples are requested to validate experimental measurements on cements (especially UCS). We use 4 samples since one cement mix allow us to generate 4 samples for shear plus 2 cubes for UCS which is normally used as reference.
- What was the load cell of the testing machine?
Answer: The load cell of the machine has a max capacity of 50000 lbf and is made by Honeywell. It has a calibration factor of 2.414 mV/V. However we have covered this in a previous paper, therefore was not mentioned in this paper again.
- During the 3 and 7 days of testing, what were the samples´ storage conditions?
Answer: all samples have been cured in water at room conditions as shown in 2.1 Sample preparation.
- What type of bond is formed between the cement and the well?
Answer: our investigations are focused on casing cement bond. This bond is caused by the cement hydration and we believe it is generated by the hydrated phse bonding with the material through the surface roughness.
- The authors mentioned that “The force is applied in small steps until the maximum applied value is reached” (lines 199-200) – what is the crosshead speed?
Answer: Thank you for your question. Ansys as numerical simulator does not have a speed as input but rather incremental steps for each a new solution is solved.
- What was the casing thickness? Do the authors think that casing thickness could affect the results?
Answer: Thank you for spotting this missing information. The casing thickness is 2.5 mm.
- How rough was the internal casing surface? Could the internal roughness affect the interface bond shear strength?
Answer: Thank you for your question. We have polished all samples with 400 grid sandpaper. We have added this information to our paper. The answer is yes, the surface roughness strongly affects the shear strength results.
Reviewer 2 Report
This is a fairly interesting study on the possible limitations of finite element analysis in terms of the analysis of cement adhesive failures.
The study is not particularly innovative and presents many criticisms:
-Line 20 Instead of talking about novelty insert the aim of this paper ... ..
-In the abstract section, in the part of the results, insert some numerical parameters obtained from the study and in the final section, instead, the possible practical implications of the study
- Line 52 indicate the author's name and then the bibliographic reference in brackets
-At the end of the introduction section, enter the purposes of the study and the null hypotheses of the same which will then have to be refuted in the light of the results obtained
- Line 73 enter the name of the author in question +
-Line 77 same speech as the previous note; also, even if briefly, indicate the sample preparation procedure
- Explicitly indicate which constraints were applied to the experimental model in the finite element analysis
-Pictures 4,5 and 7 are too large and must be merged into a single image
- A section on statistical analysis is missing and a paragraph on the limitations of the study is missing
Author Response
This is a fairly interesting study on the possible limitations of finite element analysis in terms of the analysis of cement adhesive failures. The study is not particularly innovative and presents many criticisms:
Answer: Thank you for your review. We are sorry to hear about the fact that this paper is not innovative, as we could not find any other paper fully modeling the shear bonding inside of a cylinder while comparing with experimental results.
-Line 20 Instead of talking about novelty insert the aim of this paper ... ..
Answer: Based on your recommendation we have added one more sentence discussing about the paper aim.
-In the abstract section, in the part of the results, insert some numerical parameters obtained from the study and in the final section, instead, the possible practical implications of the study
Answer: Thank you for your suggestion we have improved our abstract as per your recommendations.
- Line 52 indicate the author's name and then the bibliographic reference in brackets
Answer: Thank you for your suggestion. We added author name.
-At the end of the introduction section, enter the purposes of the study and the null hypotheses of the same which will then have to be refuted in the light of the results obtained
Answer: Thank you for your suggestion. We have added the recommended sentence.
- Line 73 enter the name of the author in question +
Answer: Thank you, corrected.
-Line 77 same speech as the previous note; also, even if briefly, indicate the sample preparation procedure
Answer: Thank you, we have tried to avoid repletion form previous papers, since most of the time it was considered similarity. We tried to shortly explain in here the process.
- Explicitly indicate which constraints were applied to the experimental model in the finite element analysis
Answer: Thank you for your comment. We have added one more sentence documented the above.
-Pictures 4,5 and 7 are too large and must be merged into a single image
Answer: Thank you for your suggestion, but we would like to keep the figures as they are, especially that this is a digital journal, and thus size of picture is irrelevant. Furthermore, we believe that condensing them will lead to poor view.
- A section on statistical analysis is missing and a paragraph on the limitations of the study is missing
Answer: Thank you for your comment, but we do not understand the need of the statistical approach in this paper. We have mentioned that we used 4 samples for each day of curing in order to extract the IBSS. 4 samples are not enough for a true statistical analysis and thus, the aim of this paper is not related to this.
We have added a new paragraph discussing the study limitations, as per your kind suggestion.
Round 2
Reviewer 2 Report
My doubts remain about the figures to be incorporated